# Epithelial–Mesenchymal Transition Expression Profile Stratifies Human Glioma into Two Distinct Tumor-Immune Subtypes

**DOI:** 10.3390/brainsci13030447

**Published:** 2023-03-05

**Authors:** Changyuan Ren, Xin Chang, Shouwei Li, Changxiang Yan, Xiaojun Fu

**Affiliations:** Sanbo Brain Hospital, Capital Medical University, No. 50, Yikesong Road, Xiangshan, Haidian District, Beijing 100093, China

**Keywords:** epithelial–mesenchymal transition, immune response, gene signature, glioma, prognosis

## Abstract

Glioma is the primary tumor with the highest incidence and the worst prognosis in the human central nervous system. Epithelial–mesenchymal transition (EMT) and immune responses are two crucial processes that contribute to it having the worst prognosis. However, a comprehensive correlation between these two processes remains elusive. The mRNA expression profiles and corresponding clinical data of patients with glioma were downloaded from public databases. EMT-related genes were collected and provided in the dbEMT database. Risk scores, Lasso regression, and enrichment analysis were conducted for functional validation. In our study, we used unsupervised clustering of EMT gene expression profiles to classify gliomas into two subtypes. We assessed the reliability of this classification system by testing it in three independent cohorts. Each subtype had different clinical and immune system characteristics. The study suggests a possible link between EMT and immune responses in gliomas.

## 1. Introduction

Gliomas, representing 81% of malignant brain tumors, are the most common tumors in the central nervous system [1]. The age-adjusted overall incidence of gliomas ranges from 4.67 to 5.73 cases per 100,000 persons (the study data are from the United States in 2014), and the overall survival (OS) time is short [2]. The 10-year survival rate for low-grade glioma is 47% with a median survival time of 11.6 years. For high-grade glioma, the median OS time of patients with grade 3 glioma is approximately 3 years, whereas those with grade 4 glioma have a poor median OS time of 15 months [3]. According to the latest Stupp protocol, the current treatment for glioma is safe and maximal tumor resection while preserving the patient’s neurological function, supplemented by concurrent chemoradiotherapy, followed by adjuvant temozolomide chemotherapy, and a series of new frontier treatment trials, such as tumor treating fields (TT fields) [4]. However, despite this detailed treatment protocol, glioma patients inevitably experience tumor recurrence or tumor progression, resulting in a poor prognosis that only further research may stand a chance of improving.

The poor prognosis for gliomas results from a combination of reasons, especially tumor heterogeneity. The heterogeneity of gliomas, encompassing intertumoral and intratumoral heterogeneity, refers to the variety of stemness renewal, angiogenesis, and metabolism capacity among patients or even within a tumor, which can result in resistance to chemoradiation therapy, leading to worse outcomes [5]. Glioma epigenetic-mediated intertumoral heterogeneity or heterogenous mutation of genes may cause the trans-differentiation from epithelial-like glioma cells into mesenchymal-like cells, leading to treatment resistance and a higher risk of recurrence [6]. This process is now widely accepted and termed epithelial–mesenchymal transition (EMT).

The development of the embryo, tissue fibrosis, wound healing, tumorigenesis, and metastasis are all significantly influenced by EMT [7,8]. The process may be triggered in a variety of cancerous tumor cells, leading to the loss of epithelial (Epi) cell characteristics such as cell–cell junctions and apical–basal polarity, and the acquisition of mesenchymal (Mes) cell characteristics that facilitate migration and finish the invasion–metastasis cascade [9]. Recent research showed that EMT transcription factors (EMT-TFs) mediate other essential and specific functions in addition to promoting cancer cell motility and dissemination [10]. EMT-TFs are important regulators of tumorigenesis, therapeutic resistance, tumor cell stemness, and tumor immune responses [11]. For instance, Snail, a typical EMT-TF, causes immunotherapy resistance in melanoma and speeds up cancer metastasis through immunosuppression by Treg cells and DCs. In addition, Twist, another important EMT-TF, can downregulate TNFα and NF-KB to mediate inflammatory suppression through type I interferons (IFNs) [6,9]. Furthermore, the Mes subtype has a dramatically greater immune cell recruitment capacity compared to its Epi counterparts [12]. These previous findings demonstrate the pivotal role of EMT in immune evasion and immunosuppression during the progression of tumors, including gliomas. However, there is currently no comprehensive evidence to systematically prove the correlation between EMT and immune responses in gliomas.

Our research involved dividing gliomas into two EMT subtypes using unsupervised clustering of EMT gene expression profiles. We then tested the reproducibility and stability of this classification system in three independent cohorts. Our analysis revealed that each of the EMT subtypes had unique clinical characteristics and tumor immune infiltration patterns. Findings from the study highlight the EMT heterogeneity of gliomas and provide a potential theoretic bridge connecting EMT and tumor immune responses, which could be a promising target for future treatment of gliomas.

## 2. Materials and Methods

### 2.1. Patients and Datasets

For our research, we analyzed gliomas using data obtained from two publicly available databases: the Chinese Glioma Genome Atlas (CGGA) and the Glioma Longitudinal Analysis (GLASS) (Appendix A). The CGGA database provided two RNA-seq datasets, as well as relevant clinical information (http://www.cgga.org.cn (accessed on 3 April 2022)). Similarly, the GLASS database contributed RNA-seq data and clinical information from 51 patients who had both initial and recurrent tumor samples available for analysis (http://synapse.org/glass (accessed on 3 January 2019)). The study was conducted according to the Helsinki Declaration and received approval from the ethics committee of Sanbo Brain Hospital. Patients’ informed consent was ensured in both public databases.

### 2.2. Identification and Validation of EMT Subtypes

To cluster EMT-related genes, we retrieved 1184 genes from the dbEMT database (dbEMT, http://dbemt.bioinfo-minzhao.org/ (accessed on 25 January 2022)). We performed Cox regression analysis on the CGGA 325 cohort to identify genes linked to overall survival (OS). We selected candidate genes with a median absolute deviation (MAD) value of at least 0.5 for consensus clustering using the “consensusClusterplus” package. We evaluated the optimal K value using the cumulative distribution function (CDF) and consensus heat map on all discovery group samples.

### 2.3. Identification of EMT-Related Signature

To identify the differentially expressed EMT genes between the two subtypes, we utilized the significance analysis of microarrays (SAM) technique through the “samr” function in R. We then used the least absolute shrinkage and selection operator (Lasso) method to identify signature genes and determine their respective coefficient (Coeff) values. Finally, we calculated the risk score for each patient in the training and validation cohorts using the following formula:$$\mathrm{Risk}\;\mathrm{score}\;=\sum_{i=1}^{n}expr\;gene\left(i\right)*Coff\;gene\left(i\right)\;.$$

### 2.4. Bioinformatic Analysis

To annotate the differential genes between EMT subtypes, we utilized gene ontology (GO) analysis for functional annotation. Additionally, we conducted gene set enrichment analysis (GSEA) to identify gene sets with statistical significance. For predicting overall survival (OS), we employed ROC curve analysis using the “pROC” R package.

### 2.5. Statistical Analysis

The statistical analyses were carried out using R language version 4.0.3 (https://cran.r-project.org/bin/windows/base/old/4.0.3/) (accessed on 25 January 2022), GraphPad Prism version 6.0 (GraphPad Inc., San Diego, CA, USA), and SPSS version 16.0 (IBM, Chicago, IL, USA). To assess the survival differences between subtypes, Kaplan–Meier analysis with a log-rank test was performed. Chi-squared tests were conducted to evaluate the differences in clinical and molecular characteristics between EMT subtypes. One-way ANOVA was used to compare the three groups. Univariate and multivariate Cox regression analyses were performed to identify prognostic factors. A *p* value less than 0.05 was considered statistically significant.

## 3. Results

### 3.1. Identification of Two Distinct EMT Subtypes

To investigate the heterogeneity of EMT in gliomas, the researchers collected 1184 EMT-related genes from a public database (dbEMT) and performed clustering analysis. A flowchart of the data resources and experimental design is shown in Figure 1A, and clinical data from the three cohorts are provided in Appendix A. Firstly, cohort1 from CGGA (CGGA cohort 325) was used as the discovery set. Cox regression analysis to identify genes associated with OS revealed 962 overlapping candidate genes between the discovery group and EMT-related genes. Then, we applied consensus clustering to the expression profiles of these candidate genes and defined two EMT subtypes, M1 and M2 (Figure 1B and Appendix A). We screened for EMT-related differentially expressed genes in CGGA cohort1. When comparing M2 to M1, 134 DEGs were obtained based on *p* < 0.05 and |Fold change| > 2 (Figure 1C, Appendix A). There was a significant prognostic difference between these two EMT subtypes (*p* < 0.001): a shorter OS was observed in M1 compared with M2 (Figure 1D). Similar grouping results were obtained with the validation groups, which were cohort2 from CGGA (CGGA cohort 693) and the GLASS cohort (Figure 1B, Appendix A).

### 3.2. Correlation of the EMT Subtypes with Clinical Features in CGGA Cohorts

Several genomic, transcriptomic, and methylation molecular features are widely accepted as basic clinical and prognostic characteristics in gliomas. A series of correlation analyses were performed using these clinical features to determine the prognostic sensitivity of the two EMT subtypes (M1 and M2) discovered in the current study. Statistical analysis (chi-squared test) revealed that in both cohorts of CGGA, a high histologic grade (WHO grade III, IV), older patients (≥42 years), and IDH wildtype were significantly associated with the M1 subtype (*p* < 0.001), while a lower histologic grade (WHO grade II), relatively younger patients (<42 years old), and IDH mutant were significantly associated with the M2 subtype (*p* < 0.001) (Figure 2). Moreover, in cohort2 of CGGA only, significant association of 1p/19q non-codeletion and recurrence were observed with the M1 subtype, while in the M2 subtype grouping, there were significantly more 1p/19q codeletion events and primary gliomas (*p* < 0.001). In addition, although not statistically different, there were more MGMT promoter unmethylated gliomas in the M1 subtype compared to the M2 subtype (*p* = 0.381 (CGGA cohort1) and 0.714 (CGGA cohort 2)). Furthermore, we also observed correlations between EMT subtypes and clinicopathological features in the GLASS cohorts (Appendix A).

### 3.3. Functional Enrichment Analysis of the EMT Subtypes

To further analyze the potential function and pathways of the obtained candidate genes, function enrichment analysis using GO and GSEA methods was performed. As shown in Figure 3A–D, GO enrichment analysis revealed top enrichment functions and pathways, including “cell activation”, “positive regulation of cytokine production”, “immune effector process”, “inflammatory response”, “regulation of defense response”, “innate immune response”, “regulation of cell activation”, “NABA MATRISOME ASSOCIATED”, “cytokine signaling in immune system”, and “neutrophil degranulation” in CGGA cohort1, and “cell activation”, “positive regulation of cytokine production”, “inflammatory response”, “innate immune response”, “positive regulation of immune response”, “regulation of cell activation”, “NABA MATRISOME ASSOCIATED”, “cytokine signaling in immune system”, “neutrophil degranulation”, and “network map of SARS-CoV-2 signaling pathway” in CGGA cohort2. GSEA analysis is another widely used enrichment method that provides accurate enrichment pathways. GSEA revealed that besides significant enrichment in function of “epithelial–mesenchymal transition”, the candidate genes were significantly enriched in pathways including “interferon GAMMA response” (NES = 3.25), “TNFA signaling via NFKB” (NES = 3.18), “KEGG lysosome” (NES = 2.80), and “cytokine–cytokine receptor interaction” (NES = 2.79) in CGGA cohort1, and “interferon GAMMA response” (NES = 2.22), “TNFA signaling via NFKB” (NES = 2.10), “KEGG lysosome” (NES = 2.09), and “antigen processing and presentation” (NES = 1.99) in CGGA cohort2. Similar results were observed in the GLASS cohort (Appendix A). To further analyze the correlation between the level of EMT and immune cells, EMT scoring was conducted to quantify the level of EMT. Most immune cells were significantly associated with the EMT score (Figure 3E). A lower EMT-scored group was enriched with a larger proportion of B cells, Tfh cells, Tgd cells, NK CD56 bright cells, DC, and mast cells, while a higher EMT-scored group had a larger proportion of Th cells, Tcm cells, Th2 cells, Th17 cells, Treg cells, CD8 T cells, NK cells, NK CD56 dim cells, iDC, aDC, eosinophils, macrophages, neutrophils, and microglia. There was no significant difference in T cells and Tem between the two EMT groups.

### 3.4. Immune Infiltration of Two EMT Subtypes Using CGGA Cohorts

The analyses above demonstrated that the candidate genes could function through active binary roles in EMT and immune-related pathways. This prompted further exploration of the potential immune activities of the candidate genes. Stromal and immune scores were calculated between subtypes using the ESTIMATE method (Figure 4A). The M1 subtype was found to have higher immune and stromal scores but lower purity compared to M2. Additionally, enrichment levels of immune cells and functions were evaluated using ssGSEA scores. As shown in Figure 4B, the M1 EMT subtype tended to have an increased level of immune cells, especially T cells, Th2 cells, Th17 cells, Treg, cytotoxic cells, NK cells, NK CD56 dim cells, iDC, aDC, eosinophils, macrophages, neutrophils, and microglia, which showed significantly different enrichment in the two CGGA cohorts. In contrast, only Tfh, Tgd, Th2, and Tem cells were enriched in the M2 EMT subtype. To validate these findings, the immune infiltration of each EMT subtype in the GLASS cohort was dissected and consistent results were obtained (Appendix A). The most enriched pathways of these genes were further evaluated. We discovered similar results in CGGA cohort2. To further explore the EMT pathway characteristics of each subtype, a total of 19 EMT-related pathways were obtained, and gene set variation analysis (GSVA) was used to quantify the enrichment of pathways. Differential analysis showed that most of the EMT pathways were enriched in the M1 subtype, such as APOPTOSIS, AXON_GUIDANCE, CHEMOKINE_SIGNALING_PATHWAY, CYTOKINE_CYTOKINE_RECEPTOR_INTERACTION, ENDOCYTOSIS, GAP_JUNCTION, etc., while the M2 subtype showed high enrichment levels of the pathways CALCIUM_SIGNALING_PATHWAY and NEUROACTIVE_LIGAND_RECEPTOR_INTERACTION (Figure 4C).

### 3.5. Identification of Immune Signature Associated with OS Using a Cox Proportional Hazards Model

Based on the EMT classification developed in this study, a prognostic signature was constructed with a Cox proportional hazards model (using R package “glmnet”). Firstly, the SAM method was used to identify 585 differentially expressed genes among CGGA cohort1 and cohort2 and the GLASS cohort, and a gene set with the best prognostic value was generated through Cox proportional hazards modeling (Figure 5A). Coefficient values and univariate Cox regression results of 11 genes were then calculated (Figure 5B,C). An analysis using the Kaplan–Meier method indicated that patients with high scores experienced notably longer overall survival (Figure 5D, *p* < 0.001). In addition, time ROC curve analyses were used to predict 1-, 3-, and 5-year OS according to risk score in CGGA cohort1, and the area under the curve (AUC) values were 0.76, 0.86, and 0.89, respectively (Figure 5E). Moreover, high scores were enriched in M2 subtype, low-grade, or IDH-mutant tumors (**** *p* < 0.0001), while no significant difference was observed between primary and recurrent gliomas (*p* = 0.0578) (Figure 5F). These observations were verified in CGGA cohort1 and the GLASS cohort and consistent results were obtained, thus validating the findings.

### 3.6. Risk Signature Is Associated with ICB Response and Immune Checkpoint

According to previous studies, combination therapy with ICB has demonstrated efficacy in preclinical models of glioma. However, the effectiveness of this therapy in patients requires further validation. To investigate the relationship between a patient’s response to ICB and their risk score, we used the Tumor Immune Dysfunction and Exclusion (TIDE) method on data from CGGA cohort1. Our analysis showed that patients in the low-risk score group (98 out of 163 or 60.1%) were more likely to respond to ICB therapy compared to those in the high-risk score group (61 out of 162 or 37.7%) (Fisher *p* < 0.0001), as demonstrated in Figure 6A. We obtained similar results when we validated the findings using CGGA cohort2 (Figure 6B) and the GLASS cohort (Appendix A) datasets. Additionally, we examined the relationship between the risk score of signatures and well-studied checkpoints. Our analysis revealed that CD276 and HAVCR2 were positively correlated with the risk score (Figure 6C). This relationship was also observed in CGGA cohort2 (Figure 6D) and the GLASS cohort (Appendix A).

## 4. Discussion

EMT is a complex process among cancers, especially gliomas [13]. EMT is closely linked to the malignancy, progression, and invasion process of gliomas, thus leading to a worse prognosis for patients [13,14]. Multiple genetic alternations and molecular events are actively involved in EMT processes. For example, ZEB-, SNAI-, MMP-, and TWIST-family members, which are key regulators of cytoskeleton rearrangement, extracellular matrix remodeling, cell adhesion contact and degradation, etc., are among the most influential activators of EMT. Master transcription factors such as β-catenin or epigenetic regulators such as miR-21, one of the major EMT-activators, can enhance extracellular matrix cleavage, subsequentially invading the surrounding microenvironment to promote glioblastoma (GBM) progression and tumor multifocality [15,16]. However, the heterogeneity of GBM means there are differences in the EMT level between different patients, and even between different cell clusters in the same patient [17,18]. This phenomenon is now considered to be one of the most important parts of tumor heterogeneity.

To date, few studies have provided comprehensive evidence regarding the problem of EMT variation in gliomas. However, advances in bioinformatic methods as well as comprehensive data collected in databases such as the Cancer Genome Atlas (TCGA), CGGA, etc., have now facilitated a thorough investigation of EMT diversity. In the current study, we first defined two subtypes of gliomas with different EMT properties and prognoses using data from the CGGA and GLASS databases, as well as genes significantly involved in the EMT pathway, as proven by research collected in the EMT database (dbEMT 2.0). We discovered that these two subgroups also presented significant differences in molecular event characteristics such as IDH mutant/wildtype and 1p/19q codeletion/non-codeletion, and significantly different clinical features such as age and WHO tumor grade. It is widely accepted that patients with lower-grade gliomas, 1p/19q codeletion, and IDH mutant gliomas have a significantly better prognosis compared with their counterparts [19,20], which is consistent with the better prognosis in the M2 subtype in the current study. Aging is another important factor in cancer prognosis, including gliomas. Clinical evidence from the Surveillance, Epidemiology, and End Results (SEER) Program showed that older glioma patients (>42 years old, consistent with our findings) had a trend of relatively worse prognosis compared with younger patients [21,22], which was not only caused by comparatively worse health conditions, but was also a result of the characteristics of the tumors themselves.

Further investigations were conducted in the current study to identify the potential characteristic(s) responsible for this prognostic diversity. We found that the differentially expressed genes between the two subgroups, which were supposed to play vital roles in EMT, were also enriched in various tumor immune-related pathways. In addition, significantly distinct immune cell enrichment capacity was observed between the two subtypes, which was consistent with the findings of the distinct distribution of immune cells between EMT scoring levels. Different EMT scoring systems have been established in previous studies as promising and convincing tools with which to monitor the role of EMT in cancer progression and therapeutic resistance. In the present study, most of the immune cells exhibited distributional differences between the M1 and M2 subgroups, as well as according to EMT levels. For instance, the M1 subgroup had significantly higher stromal and immune scores and lower tumor purity compared to the M2 subgroup. Furthermore, Th cells, Tcm cells, Th2 cells, Th17 cells, Treg cells, CD8 T cells, NK cells, NK CD56 dim cells, iDC, aDC, eosinophils, macrophages, neutrophils, and microglia tended to be positively associated with EMT levels, while, conversely, B cells, Tfh cells, Tgd cells, NK CD56 bright cells, DC, mast cells, Tfh, Tgd, Th2, and Tem were relatively negatively enriched in accordance with EMT levels. Tumor-associated immune cells, which have crucial roles within the tumor microenvironment (TME), were discovered to function as key regulators and effectors of EMT [23,24]. Within the tumor microenvironment (TME), immune cells can secrete various factors, such as cytokines and chemokines, which can influence the epithelial–mesenchymal transition (EMT) process in cancer cells through multiple pathways. Cancer cells, in turn, can communicate with immune cells, leading to cell plasticity and the release of immunosuppressive substances. This interaction can create an immunosuppressive microenvironment that promotes the EMT process and contributes to tumor invasion and metastasis [12,25]. For example, stromal cells in the TME, including macrophages, fibroblasts, myeloid-derived suppressor cells (MDSCs), T cells, B cells, adipocytes, mast cells, and other stromal cells, act like “partners in crime” in cancer progression [13,26]. However, until now, there has been very limited research that could provide evidence to quantify this correlation between the immune response and EMT in gliomas. Our results present a very detailed and comprehensive correlation between EMT and immune cells. However, the results do not necessarily mean that these immune cells directly promote or inhibit the EMT process, though they indicate that these cells should actively affect EMT. These intriguing results need to be further elucidated.

The treatment of glioma has currently reached an obvious bottleneck. Alkylating agents such as temozolomide are still commonly used in the adjuvant treatment of glioma. Immunotherapies targeting PD1/PD-L1, etc., which are recognized in research on other tumors, have not shown marked effectiveness in the field of gliomas [27]. Therefore, there is an urgent need to identify effective new therapeutic drugs in glioma-related research. Targeting EMT is a potential new treatment method that is also believed to be closely related to tumor immunity. However, the precise molecular mechanisms that could be actively involved in both the EMT process and tumor immune escape remain elusive. In the current study, we developed a risk model using a Lasso regression method. Eleven candidate genes were also listed that may represent the molecular bridge connecting EMT and the immune response.

## 5. Conclusions

In conclusion, our study identified two EMT subtypes that exhibited unique immune characteristics. The results suggest that EMT plays a crucial role in immune evasion and immunosuppression in glioma progression, and our findings may serve as novel predictors and potential therapeutic targets for the treatment of tumors.

## Figures and Tables

**Figure 1 brainsci-13-00447-f001:**
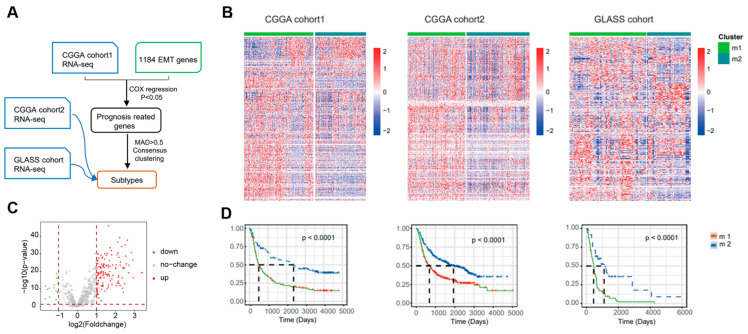
Identification of distinct epithelial–mesenchymal transition subtypes in gliomas through EMT gene profiling. (**A**) Flow chart shows the data resources and total experimental design. CGGA cohort2 and GLASS cohort were collected as validation sets. MAD, median absolute deviation. (**B**) Heatmap of two EMT subtypes defined in three cohorts. (**C**) Volcano plot of the differentially expressed genes (DEGs) by comparing M1 to M2 from CGGA cohort1. Venn diagram for the EMT−related differentially genes. *p* < 0.05, |FC| > 2. (**D**) In three different groups, survival analyses revealed notable distinctions between the two EMT subtypes. To determine the statistical significance between the subtypes, the log-rank test was used to calculate the *p* value.

**Figure 2 brainsci-13-00447-f002:**
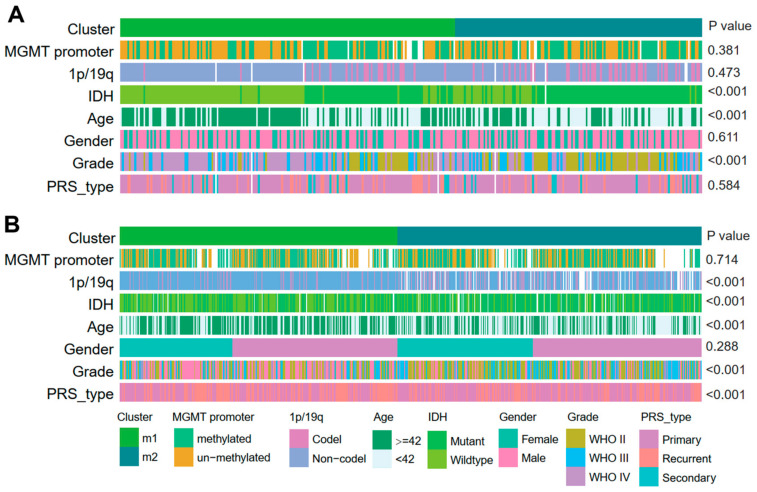
Clinical characteristics of EMT subtypes in CGGA cohorts. (**A**,**B**) Correlation of our classification with clinical characteristics and previous subclasses in CGGA cohorts.

**Figure 3 brainsci-13-00447-f003:**
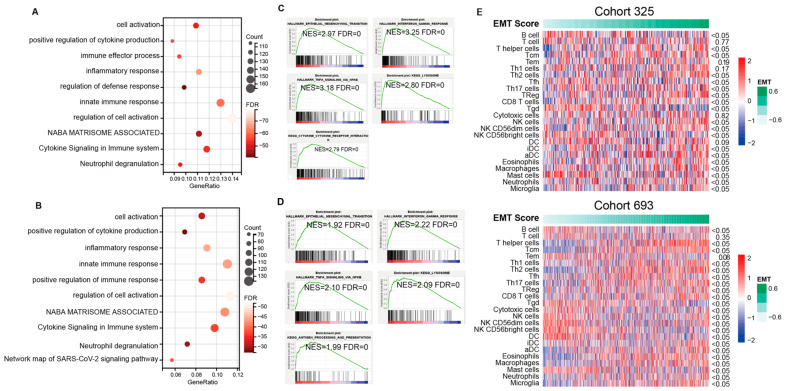
A detailed correlation between EMT and glioma immune responses. (**A**–**D**) Enrichment pathways generated by GO and GSEA method using the overlapped genes showed the top potential functions in cohort1 and cohort2 of CGGA. GO analysis of upregulated genes in M1 subtype (CGGA cohort1 (**A**), CGGA cohort2 (**C**)). Enriched functions of M1 subtype identified through GSEA (CGGA cohort1 (**B**), CGGA cohort2 (**D**)). (**E**) Correlation analysis shows the association between immune cells and EMT levels.

**Figure 4 brainsci-13-00447-f004:**
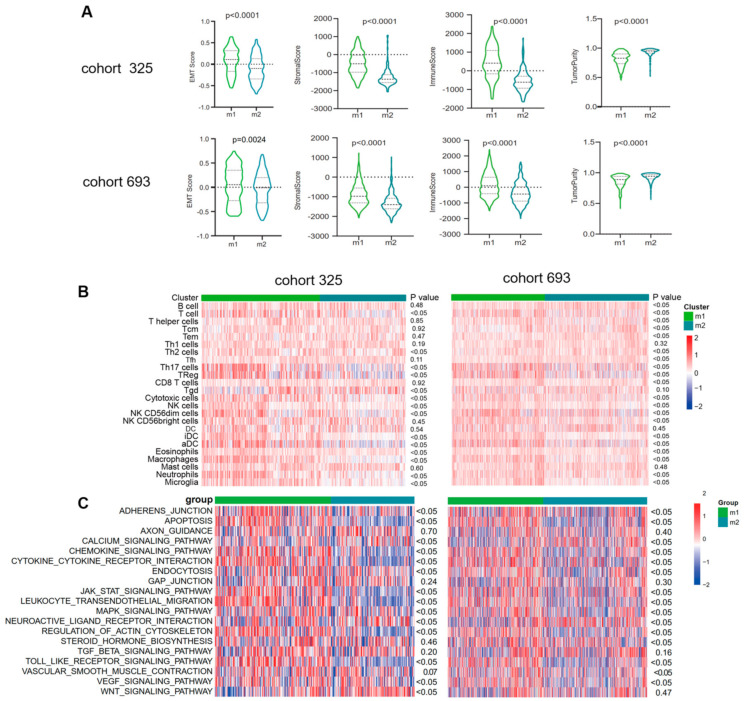
Functional enrichment analysis of the EMT subtypes showed close correlation between EMT and immune−related responses. (**A**) Violin plots showing scores for immune, stromal, and tumor purity in various EMT subtypes in the CGGA cohorts (ANOVA test). (**B**) Heatmaps show differential enrichments of immune−related cells between two EMT subtypes. (**C**) Heatmap showing the EMT−related pathways differentially enriched in the m1 versus m2. ANOVA test was used for statistical analysis, and the *p* values were labeled.

**Figure 5 brainsci-13-00447-f005:**
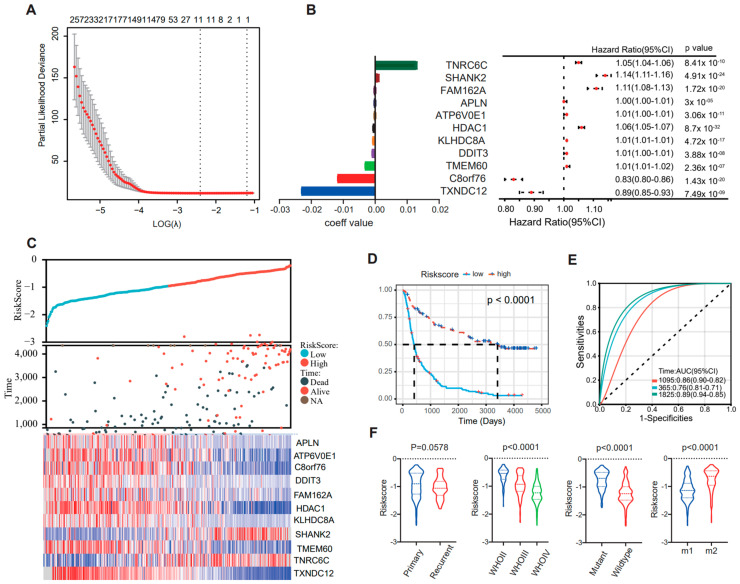
The Cox proportional hazards model was used to identify an EMT signature that is correlated with overall survival. (**A**) Cross-validation for tuning parameter selection in the proportional hazards model. (**B**) Forrest map shows the expression levels of 11 signature genes. (**C**) Distribution of the risk score, overall survival (OS), and expression level of 11 genes in the risk signature. (**D**) Kaplan–Meier survival analysis of the EMT signature in patients with gliomas. The *p* value was calculated using the log-rank test. (**E**) Time receiver operating characteristic (ROC) curve analyses to predict 1−, 3−, and 5−year OS according to risk score in CGGA cohort1 datasets. (**F**) Distribution of the risk score in glioma patients stratified by primary/recurrence, IDH mutant/wildtype, or WHO grade.

**Figure 6 brainsci-13-00447-f006:**
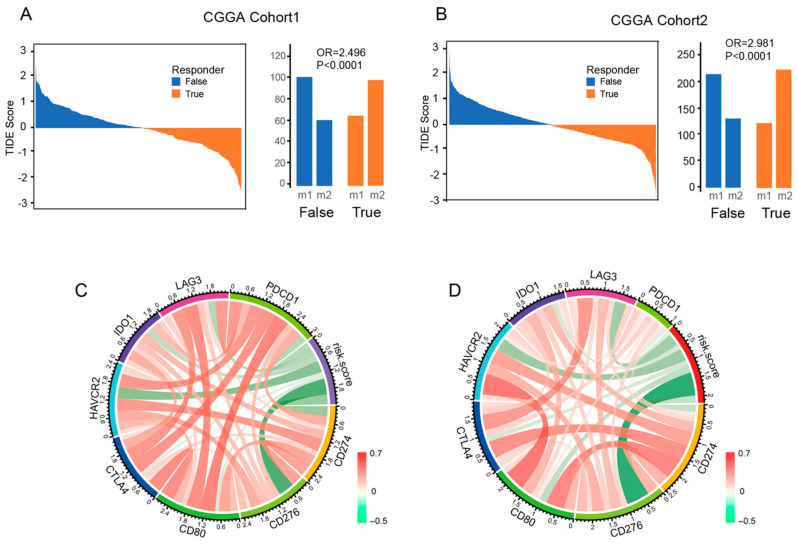
The risk signature is associated with the ICB response and immune checkpoint. (**A**,**B**) The TIDE score and response results to immunotherapy of patients with glioma. (**C**,**D**) The correlation coefficient between risk score and immune checkpoints.

## Data Availability

Three datasets were used in this study from two databases: CGGA and GLASS (Appendix A). The data from the CGGA database contain two RNA-seq datasets, as well as the relevant clinical information (http://www.cgga.org.cn, accessed on 17 October 2022). In addition, RNA-seq data and clinical information for 51 patients with initial and recurrent tumor samples (Appendix A) were gathered from the glioma longitudinal analysis (GLASS) consortium (http://synapse.org/glass, accessed on 17 October 2022). This study was carried out in accordance with the Declaration of Helsinki and approved by the ethics committee of Sanbo Brain Hospital, and patients’ informed consent was ensured in the two public databases.

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
