# Peer review of "Epithelial–Mesenchymal Transition Expression Profile Stratifies Human Glioma into Two Distinct Tumor-Immune Subtypes"

_brainsci, 2023, doi:10.3390/brainsci13030447_

Round 1

Reviewer 1 Report

The paper is good. It can be accepted for publication.

The paper is good. It can be accepted for publications.

e accepted for publications.

Author Response

Thank you very much for your comments and we really appreciate your consideration of our manuscript.

Reviewer 2 Report

The study investigated the mRNA expression profiles in gliomas with regard to the epithelial-mesenchymal transition (EMT). Clustering analysis including EMT-related genes identified two distinct groups, termed as M1 and M2. These distinct groups revealed differences in their overall survival. The EMT subtypes were associated with clinical features, such as the histological grading of the tumor, the patients’ age and the IDH mutation status and the 1p/19q codeletion/non-codeletion. Enrichment analysis revealed the involvement of several immune-related pathways by comparing differentially expressed genes between M1 and M2.

Considering the importance of prognostic markers for patients with glioma, this article is of interest to the scientific community of Brain Sciences. I would suggest the acceptance of this article after  a careful revision and the clarification of the points below:

- How do this study's findings correlate with publicly available data from other studies (e.g. TCGA)?

- The formula for the risk score in line 99-100 should be revised.

- Figure 1: Add a more detailed figure caption.

- The nomenclature of the WHO grading system (WHO CNS 2016) states the grades as follows: WHO grade I; WHO grade II; WHO grade III; and WHO grade IV. Adapt in line 139-140, line 142 and in Figure 2.

- Figure 3: In the results section it is stated that the GO enrichment terms and the GSEA results are shown. Why is KEGG listed in the figure caption? Add a more detailed figure caption. A consistent labeling of the cohorts throughout the manuscript would be preferable.

- Figure 4: Add a more detailed figure caption. A consistent labeling of the cohorts throughout the manuscript would be preferable.

- In line 222, a comma is missing between “AXON_GUIDANCE” and “CHEMOKINE_SIGNALING_PATHWAY”.

- Figure 5: Subfigure 5A is not mentioned in the results section. In line 239, line 240 and line 241 Figure 6B, 6C and 6D are mentioned. Most probably Figure 5 is intended. Subfigure 5E and 5F are not mentioned in the results section.

Reviewer 3 Report

I have no specific comments to authors.

Author Response

Thank you. It is really very kind of you to consider the publication of manuscript.

Reviewer 4 Report

The objective of this manuscript is to identify two EMT subtypes based on unsupervised  clustering of EMT gene expression profiles. Furthermore, the authors analyzed each of the two EMT subtypes was associated with distinct clinical characteristics and tumor immune infiltration. This is a good idea. However, the authors divided two types of EMT based on 1184 genes. There were too many genes in each group. It is not helpful for further understanding the relationship between EMT and immune response. I will buy this concept if the authors first construct the prognostic signature with a Cox proportional model in Fig.5, then, the author can further analyze Clinicopathological Feature, evaluate Survival Prediction Nomogram, and analyze the distinction of tumor immune microenvironment and underlying implications for Immunotherapy in Low and High risk groups.

1.The relationship between EMT and immune response has been reported e.g.  PMCID: PMC8219790 .Please try to dig more interesting findings.

2.In figure 1, the authors divide two types of EMT by 1184 genes. please show how much and what genes belong to M1 and M2, respectively. By the way, I could not find the supplementary data.

3.It is more valuable if the authors could analyze the therapeutic effect of checkpoint blockage among different EMT types.

4.The authors should validate the EMT signature in patient samples by IHC, LCM(Laser capture microdissection),or other methods.

Round 2

Reviewer 2 Report

The authors addressed all comments. I would recommend the acceptance of this manuscript for publishing in the current revised version.

Author Response

Thank you again for your comments. 

Reviewer 4 Report

The revised version has been significantly improved according to the comments. I would like to recommend the manuscript for publication.